# Microbial Interactions as a Sustainable Tool for Enhancing PGPR Antagonism against Phytopathogenic Fungi

Ana M. Santos [1], Ana Soares [1,2], João Luz [3], Carlos Cordeiro [3], Marta Sousa Silva [3], Teresa Dias [1], Juliana Melo [1], Cristina Cruz [1,*] and Luís Carvalho [1,2]

1   cE3c—Centre for Ecology, Evolution and Environmental Changes & Change—Global Change and Sustainability Institute, Faculdade de Ciências, Universidade de Lisboa, Campo Grande, Bloco C2, 1749-016 Lisboa, Portugal; amhce.santos@gmail.com (A.M.S.); ana.filipa.soares8@gmail.com (A.S.); mtdias@fc.ul.pt (T.D.); jmdconceicao@fc.ul.pt (J.M.); luis.carvalho@bioscale.pt (L.C.)
2   BioScale, Rua Nova da CEE, 2005-008 Santarém, Portugal
3   Laboratório de FTICR e Espectrometria de Massa Estrutural, Departamento de Química e Bioquímica, Biosystems and Integrative Sciences Institute (BioISI), Faculdade de Ciências, Universidade de Lisboa, 1749-016 Lisboa, Portugal; cacordeiro@ciencias.ulisboa.pt (C.C.); mfsilva@ciencias.ulisboa.pt (M.S.S.)
*   Correspondence: ccruz@fc.ul.pt; Tel.: +351-964-880-356

**Abstract:** Microbial interactions, which regulate the dynamics of eco- and agrosystems, can be harnessed to enhance antagonism against phytopathogenic fungi in agriculture. This study tests the hypothesis that plant growth-promoting rhizobacteria (PGPR) can also be potential biological control agents (BCAs). Antifungal activity assays against potentially phytopathogenic fungi were caried out using cultures and cell-free filtrates of nine PGPR strains previously isolated from agricultural soils. Cultures of *Bacillus* sp. BS36 inhibited the growth of *Alternaria* sp. AF12 and *Fusarium* sp. AF68 by 74 and 65%, respectively. Cell-free filtrates of the same strain also inhibited the growth of both fungi by 54 and 14%, respectively. Furthermore, the co-cultivation of *Bacillus* sp. BS36 with *Pseudomonas* sp. BS95 and the target fungi improved their antifungal activity. A subsequent metabolomic analysis using Fourier-transform ion cyclotron resonance mass spectrometry (FTICR-MS) identified fengycin- and surfactin-like lipopeptides (LPs) in the *Bacillus* sp. BS36 cell-free filtrates, which could explain their antifungal activity. The co-production of multiple families of LPs by *Bacillus* sp. BS36 is an interesting feature with potential practical applications. These results highlight the potential of the PGPR strain *Bacillus* sp. BS36 to work as a BCA and the need for more integrative approaches to develop biocontrol tools more accessible and adoptable by farmers.

**Keywords:** *Alternaria* sp.; *Bacillus*; biocontrol; co-culture; Fourier-transform ion cyclotron resonance mass spectrometry; *Fusarium* sp.; lipopeptides

## 1. Introduction

Plant diseases cause major economic losses and pose threats to food and environmental safety. Phytopathogenic fungi are responsible for 70 to 80% of plant diseases, negatively affecting crop growth and yield [1]. The current management of plant diseases comprises an overuse of synthetic fungicides, which has several undesirable effects on the environment, including on beneficial microorganisms and human health [2,3]. Over recent years, limitations on the use of pesticides in European agriculture have been implemented. For instance, the EU Farm to Fork strategy aims to reduce the use of synthetic pesticides on crops by 50% by 2030 [4]. In this context, the use of plant growth-promoting rhizobacteria (PGPR) for biological control of phytopathogenic fungi is a valuable and sustainable alternative to the use of synthetic fungicides [5].

The antagonistic action of PGPR often relies on their production of bioactive metabolites. *Pseudomonas* and *Bacillus* are commonly used as biological control agents (BCAs) due to their ability to produce metabolites with different structures and broad-spectrum

antifungal activity [6]. *Pseudomonas* species are able to produce a wide array of antifungal metabolites, including 2,4-diacetylphloroglucinol [7,8], cell-wall-degrading enzymes (CWDEs) [9,10], phenazines [11,12], pyoluteorin [8], pyrrolnitrin [13,14], and volatile organic compounds (VOCs) [15,16]. *Bacillus* species also produce important antifungal compounds, including CWDEs [17,18] and VOCs [19,20]. In addition, both genera are known for their ability to produce lipopeptides (LPs) [21–25]. LPs are amphiphilic secondary metabolites with a low molecular weight synthesised by multi-enzyme complexes known as non-ribosomal peptide synthetases (NRPSs). *Pseudomonas* and *Bacillus* species produce a variety of LPs with different structural characteristics. These differences comprise variations in the amino acids of the peptide domain and variations in the length and composition of the lipid tail [26]. *Bacillus* is the most studied genus capable of secreting LPs [27]. *Bacillus* produces LPs that are commonly classified into three main families: fengycin [28], iturin [29], and surfactin [30]. Compounds from different families vary in their structure and therefore in their antimicrobial properties [31]. Fengycin and iturin are the main LPs with strong antifungal activity against phytopathogens [28,32–34].

In nature, the biosynthesis and accumulation of natural products, such as LPs and other metabolites, are triggered by interactions between microorganisms, such as competition for nutrients and space. However, certain biosynthetic pathways responsible for producing secondary metabolites are silenced in laboratory conditions where axenic cultures are maintained [35]. Thus, mimicking the natural microbial environment using the mixed fermentation of different microorganisms (co-cultivation) promotes microbial interactions. This can increase the accumulation of certain metabolites or activate the expression of silent biosynthetic gene clusters, leading to the production of new metabolites [36]. Therefore, microbial co-cultivation has been used to elicit the synthesis of antimicrobial secondary metabolites in an attempt to increase antifungal activity [36–38].

The aims of this study were to (1) compare a set of PGPR strains for their ability to supress the growth of potentially phytopathogenic fungi in an attempt to select suitable candidates for BCAs; (2) establish microbial interactions through co-cultivation as an approach to increasing the production of existing and new metabolites and enhancing the antifungal activity of the PGPR; and (3) determine the nature and stability of extracellular metabolites with antimicrobial activity.

## 2. Materials and Methods

### 2.1. PGPR Strains

Nine PGPR strains, previously isolated from agricultural soils and characterised for their plant growth-promoting potential, were obtained from a culture collection maintained in the company Bioscale (Santarém, Portugal). The strains had been stored in Nutrient Broth (NB, Biokar Diagnostics, Allonne, France) supplemented with 20% ($w/v$) glycerol at $-80$ °C and were revitalized on Nutrient Agar (NA, Biokar Diagnostics, Allonne, France) at 28 °C for 24 h or until visible growth. The incubation time in NB was optimised to reach a minimum concentration of $10^7$ CFU mL$^{-1}$. For the antifungal activity assays, the strains were sub-cultured on NB at 28 °C, 160 rpm, for 24–48 h, depending on the PGPR strain, with the initial concentration set to $10^6$ CFU mL$^{-1}$.

### 2.2. Fungal Strains

*Alternaria* sp. AF12 and *Fusarium* sp. AF68 were obtained from a fungal collection maintained in Bioscale (Santarém, Portugal). Four additional fungal strains, *Alternaria* sp. FP3, *Botrytis* sp. SM-D1, *Fusarium* sp. SM-D3, and *Stemphylium* sp. FP5, were recently isolated from leaves of *Pyrus communis* L. cv. "Rocha" with brown spots. All fungal strains had been stored in glycerol 10% ($w/v$) at $-80$ °C and were grown on Potato Dextrose Agar (PDA, Biokar Diagnostics, Allonne, France) at 28 °C for 5–6 days prior to their use in the antifungal activity assays. Fungal strains were identified based on conidia morphology and ITS sequence analysis.

### 2.3. Genomic DNA Extraction, PCR, and Phylogenetic Analysis of PGPR Strains

To identify the PGPR strains, the genomic DNA (gDNA) of the bacterial strains was extracted following a modified and optimised version of the guanidium thiocyanate method described by Pitcher et al. [39]. The partial 16S rRNA gene was amplified using the forward primer PA/8F (5′-AGAGTTTGATCCTGGCTCAG-3′) [40,41] and the reverse primers 907R (5′-CCGTCAATTCMTTTRAGTTT-3′) [42] or 1392R (5′-ACGGGCGGTGTGTRC-3′) [43]. PCR reactions were performed in a Biometra Uno II thermal cycler (Biometra GmbH, Göttingen, Germany) under the following conditions: initial denaturation at 94 °C for 3 min, followed by 35 cycles at 94 °C for 1 min, 55 °C for 1 min and 72 °C for 1 min, and final extension at 72 °C for 3 min. The amplified products were sequenced by Eurofins Genomics (Ebersberg, Germany) and the newly generated 16S rRNA gene sequences were blasted against GenBank. Maximum likelihood (ML) trees were constructed using MEGA X (v. 10.2.6) and the General Time Reversible (GTR) nucleotide substitution model. The best-scoring ML tree was estimated by conducting a bootstrap analysis of 1000 replicates.

### 2.4. Antifungal Activity in Dual Culture Assay

The antifungal activity of the PGPR strains was tested using in vitro dual culture assays in Petri plates (60 mm diameter) containing 10 mL of PDA. A mycelial disc with a 6 mm diameter from the target fungal strain was inoculated on the centre of the Petri plate, whereas four aliquots (5 µL) of the fresh bacterial monoculture were inoculated on the periphery. Control plates were prepared using sterile ultrapure water aliquots. After incubation at 28 °C for 6 days, the radial fungal growth was measured in the control and treated plates. Mycelial growth inhibition (MGI, in%) was expressed as the radial reduction in the fungal mycelium observed in the treated plates compared to the corresponding control plates, and was calculated using the following equation:

$$\text{MGI}\,(\%) \;=\; \left[ \frac{(R_1 - R_2)}{R_1} \right] * 100 \tag{1}$$

where $R_1$ is the radial growth of the fungal mycelium in the control plates and $R_2$ is the radial growth of the fungal mycelium in the treated plates. The experiment was carried out with four replicates for the control and for each bacterium–fungus combination.

### 2.5. Antifungal Activity of Extracellular Metabolites in Cell-Free Filtrates

Bacterial cells were removed by centrifugation of the fresh bacterial monoculture (after 24–48 h, as described in Section 2.1) three times at 3220× *g* for 15 min at 4 °C. The supernatant was filtered through a 0.22 µm pore size membrane (Merck Millipore, Darmstadt, Germany). The resulting cell-free filtrate was added to PDA at a final concentration of 10% (*v/v*) (1 mL of bacterial filtrate and 9 mL of PDA). A mycelial disc with a 6 mm diameter from the target fungal strain was inoculated on the centre of the Petri plate. Control plates were prepared using PDA with sterile ultrapure water. The plates were incubated at 28 °C for 6 days. The radial fungal growth was measured in the control and treated plates, and the MGI was determined as described above (see Section 2.4). The experiment was carried out with four replicates for the control and for each bacterium–fungus combination.

### 2.6. Bacterial Co-Cultivation including *Bacillus* sp. *BS36*

To increase the antifungal activity of the cell-free filtrates, *Bacillus* sp. BS36 was co-cultured with additional PGPR strains. A total of eight pairwise interactions were tested. From fresh bacterial subcultures, each pair was inoculated on NB, with the initial inoculum of each strain set at $5 \times 10^5$ CFU mL$^{-1}$ (1:1). The cultures were incubated at 28 °C and 160 rpm for 48 h. The antifungal activity of the cultures' filtrates against *Alternaria* sp. AF12 and *Fusarium* sp. AF68 was determined as described above (see Section 2.4).

### 2.7. Co-Cultivation of Bacillus sp. BS36 and Inactivated Fungal Cells

To increase the antifungal activity of the cell-free filtrates, *Bacillus* sp. BS36 was cultivated in the presence of thermally inactivated cells of the target microorganisms, *Alternaria* sp. AF12 and *Fusarium* sp. AF68. Fungal strains were grown on Brain Heart Infusion (BHI, Biokar Diagnostics, Allonne, France) at 28 °C and 160 rpm for 3 days. The cultures were centrifuged three times at $12,000 \times g$ for 20 min at 4 °C. Each pellet was resuspended in an equal volume of NB and autoclaved at 121 °C for 30 min. The resulting suspension was added to NB at a final concentration of 10% (*v/v*), and *Bacillus* sp. BS36 was inoculated with an initial concentration set to $10^6$ CFU mL$^{-1}$. The cultures were incubated at 28 °C and 160 rpm for 48 h. The antifungal activity of the cultures' filtrates against *Alternaria* sp. AF12 and *Fusarium* sp. AF68 was determined as described above (see Section 2.4).

### 2.8. Effects of Proteinase K Treatment and Heating on Antifungal Activity of Bacillus sp. BS36 Cell-Free Filtrates

To evaluate the stability of the antifungal metabolites produced by *Bacillus* sp. BS36, the cell-free filtrates were exposed to physicochemical treatments, with 0.1 mg mL$^{-1}$ proteinase K (Invitrogen, Carlsbad, CA, USA) at 37 °C for 60 min or incubated at 80 °C for 30 min, before being used for antifungal activity assays. The antifungal activity against *Alternaria* sp. AF12 and *Fusarium* sp. AF68 was determined as described above (see Section 2.4).

### 2.9. Antifungal Activity of Crude Extracts of LPs from Bacillus sp. BS36

The antifungal activity of the LPs present in the *Bacillus* sp. BS36 cell-free filtrates was investigated. The LPs were extracted by acid precipitation and solvent extraction. The filtrates were precipitated by adjusting the pH to 2.0 with concentrated HCl and were stored overnight at 4 °C. The precipitates were collected by centrifugation at $12,000 \times g$ for 30 min at 4 °C and extracted with methanol at room temperature and 160 rpm. The solution was filtered through Whatman filter paper No. 1 (Whatman, London, UK) and dried with a rotary vacuum evaporator SyncorePlus (Buchim, Flawil, Switzerland) at 56 °C, 278 bar, 160 rpm, for 3–4 h. The remaining extract of the LPs was dissolved in $1\times$ PBS buffer (pH 7.3–7.4) (Invitrogen, Carlsbad, CA, USA) and filtered through a 0.22 μm pore size membrane (Merck Millipore, Darmstadt, Germany). The antifungal activity against *Alternaria* sp. AF12 and *Fusarium* sp. AF68 was determined as described above (see Section 2.4).

### 2.10. FTICR-MS and Data Analysis

The presence of LPs in the cell-free filtrates of *Bacillus* sp. BS36 was screened following an untargeted metabolomics approach using Fourier-transform ion cyclotron resonance mass spectrometry (FTICR-MS). Two samples were prepared by diluting the filtrate 2-fold in methanol (500 μL of filtrate in 500 μL of methanol). For internal mass spectrum calibration, human leucine enkephalin was added ($[M + H]^+$ = 556.27657 Da) at a concentration of 0.1 μg mL$^{-1}$. Formic acid (MS grade, Sigma-Aldrich, St. Louis, MO, USA) was also added to the samples at a final concentration of 0.1% (*v/v*). The samples were ionized by electrospray ionization in positive mode (ESI$^+$) and spectra were acquired between 200 and 1500 $m/z$. The mass spectra were analysed using the software package Data Analysis 5.0 and MetaboScape 5.0 (Brüker Daltonics, Bremen, Germany), considering peaks with a minimum signal-to-noise ratio of 4. The spectra were aligned and compound identification was performed using the Human Metabolome Database (HMDB, from 27 May 2022) [44] and LOTUS (from 16 September 2022) [45], uploaded to MetaboScape 5.0, considering the adducts H$^+$, Na$^+$, and K$^+$, and a mass deviation of less than 1 ppm.

### 2.11. Statistical Analysis

The MGI values are presented as the means of four independent replicates with standard deviation (SD). The data were analysed statistically by one-way ANOVA, the

Tukey post hoc test, and the independent samples *t*-test using the software IBM SPSS Statistics (v. 29).

## 3. Results

### 3.1. PGPR Strains Identification

In this study, we amplified the 16S rRNA gene of the nine PGPR strains and an ML analysis was generated from 38 aligned sequences (Figure 1). Based on these molecular data, the PGPR strains were clustered with three different genera, including *Bacillus* (BS36 and BS84), *Priestia* (BS1 and BS90), and *Pseudomonas* (BS2, BS3, BS27, BS94, and BS95), with moderate or high bootstrap values (Figure 1).

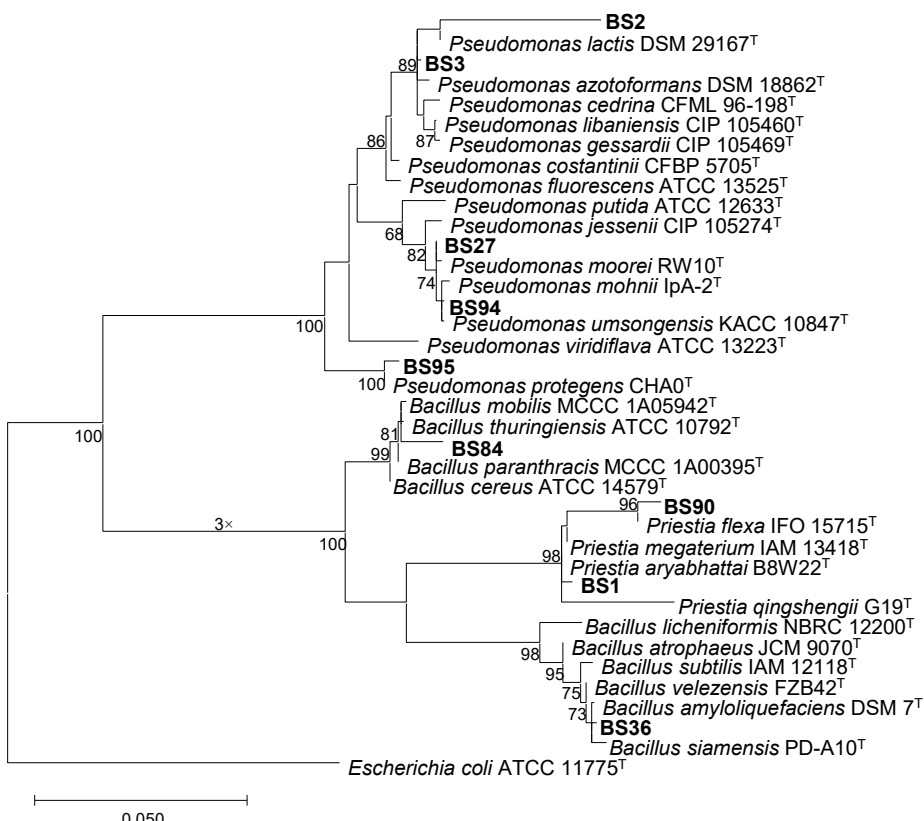

**Figure 1.** Maximum likelihood (ML) phylogenetic tree based on partial sequences of 16S rRNA gene. Bootstrap values >70% are shown at the nodes. Type strains are noted with T superscript ($^T$). The strains from this study are indicated in bold font. The scale bars represent the expected number of nucleotide changes per site. *Escherichia coli* ATCC 11,775 strain was used as outgroup.

### 3.2. Screening of Antifungal Activity of PGPR Strains

The nine PGPR strains showed a wide range of MGI against both potentially phytopathogenic fungi, *Alternaria* sp. AF12 and *Fusarium* sp. AF68 (Table 1). The antifungal activity against *Alternaria* sp. AF12 varied from 0 to 74.0% (higher for *Bacillus* sp. BS36), and against *Fusarium* sp. AF68, varied from 0 to 65.4% (higher for *Bacillus* sp. BS36). *Bacillus* sp. BS36 (74.0 and 65.4%), *Pseudomonas* sp. BS95 (38.5 and 61.8%), and *Pseudomonas* sp. BS2 (20.2 and 7.8%) were the most efficient strains in the MGI of *Alternaria* sp. AF12 and *Fusarium* sp. AF68, respectively. *Pseudomonas* sp. BS94 also slightly inhibited the growth of both fungi. All *Pseudomonas* (BS2, BS3, BS27, BS94, and BS95) strains were able to inhibit the growth of *Fusarium* sp. AF68. The *Priestia* (BS1 and BS90) strains were the only ones that did not inhibit the growth of any of the fungal strains (Table 1).

**Table 1.** Antifungal activity of PGPR strains against *Alternaria* sp. AF12 and *Fusarium* sp. AF68, expressed by MGI.

| PGPR Strains | | Mycelial Growth Inhibition (MGI,%) [1] | |
|---|---|---|---|
| | | *Alternaria* sp. AF12 | *Fusarium* sp. AF68 |
| *Bacillus* sp. | BS36 | 74.0 ± 3.2 | 65.4 ± 3.0 |
| | BS84 | 3.9 ± 2.7 | 0.0 ± 0.0 |
| *Priestia* sp. | BS1 | 0.0 ± 0.0 | 0.0 ± 0.0 |
| | BS90 | 0.0 ± 0.0 | 0.0 ± 0.0 |
| *Pseudomonas* sp. | BS2 | 20.2 ± 1.7 | 7.8 ± 4.4 |
| | BS3 | 0.0 ± 0.0 | 7.8 ± 4.4 |
| | BS27 | 0.0 ± 0.0 | 6.9 ± 1.7 |
| | BS94 | 2.9 ± 1.7 | 7.8 ± 2.0 |
| | BS95 | 38.5 ± 6.1 | 61.8 ± 1.7 |

[1] Values represent the mean ± SD (n = 4).

*Bacillus* sp. BS36 and *Pseudomonas* sp. BS95 were selected for additional studies of antifungal activity against four potentially phytopathogenic fungi recently isolated from *Pyrus communis* L. cv. "Rocha" brown spots (Figure 2). *Bacillus* sp. BS36 revealed the highest MGI of *Alternaria* sp. FP3 (82.7 ± 4.3%), *Botrytis* sp. SM-D1 (60.6 ± 1.7%), *Fusarium* sp. SM-D3 (69.2 ± 0.0%), and *Stemphylium* sp. FP5 (76.9 ± 0.0%). *Pseudomonas* sp. BS95 only inhibited the growth of *Alternaria* sp. FP3 (38.5 ± 4.7%) and *Stemphylium* sp. FP5 (29.8 ± 1.7%) (Figure 2).

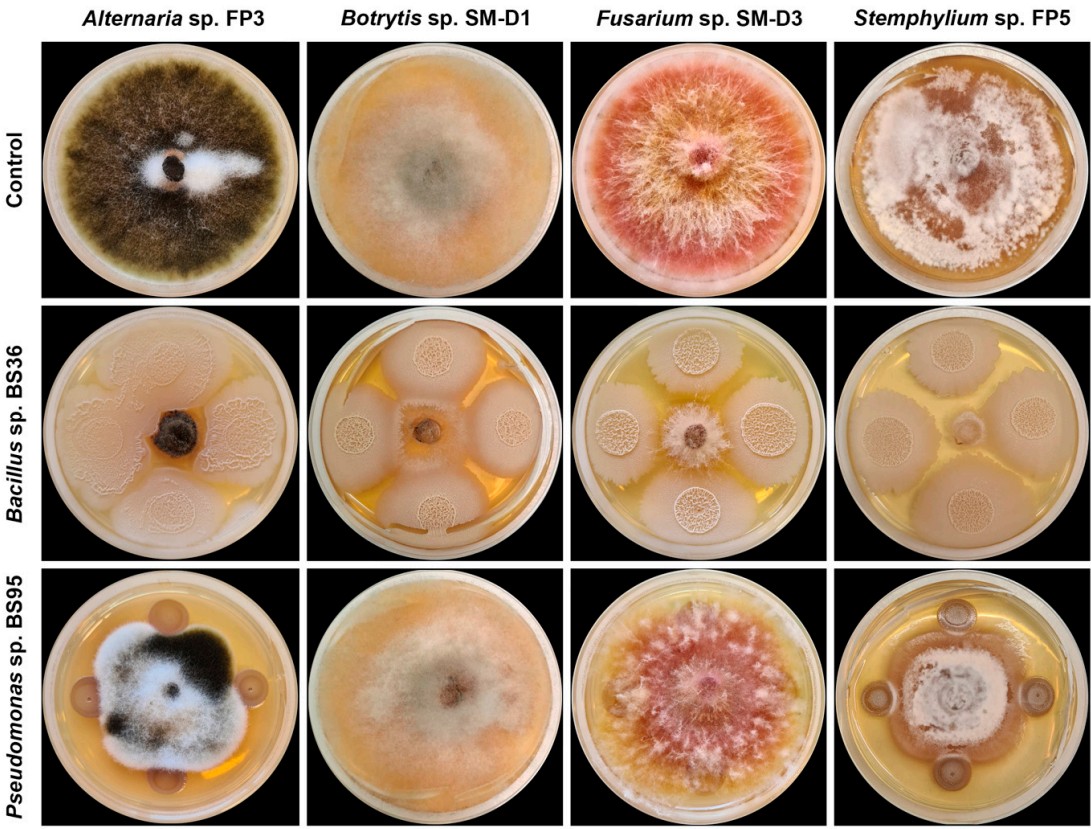

**Figure 2.** Antifungal activity of *Bacillus* sp. BS36 and *Pseudomonas* sp. BS95 against *Alternaria* sp. FP3, *Botrytis* sp. SM-D1, *Fusarium* sp. SM-D3, and *Stemphylium* sp. FP5, in dual culture assay. *Bacillus* sp. BS36 and *Pseudomonas* sp. BS95, respectively: MGI = 82.7 ± 4.3 and 38.5 ± 4.7% for *Alternaria* sp. FP3, MGI = 60.6 ± 1.7 and 0.0 ± 0.0% for *Botrytis* sp. SM-D1, MGI = 69.2 ± 0.0 and 0.0 ± 0.0% for *Fusarium* sp. SM-D3, MGI = 79.6 ± 0.0 and 29.8 ± 1.7% for *Stemphylium* sp. FP5.

### 3.3. Antifungal Activity of Extracellular Metabolites in Cell-Free Filtrates

The cell-free filtrates derived from cultures of *Bacillus* sp. BS36 and *Pseudomonas* sp. BS95 were also tested for their antifungal activity against *Alternaria* sp. AF12 and *Fusarium* sp. AF68 (Figure 3). The *Bacillus* sp. BS36 filtrates inhibited the growth of both fungal strains (53.9% MGI for *Alternaria* sp. AF12 and 14.4% MGI for *Fusarium* sp. AF68). However, the *Pseudomonas* sp. BS95 filtrates only inhibited the growth of *Alternaria* sp. AF12 (5.8%). The MGI induced by the cell cultures of both PGPR strains was significantly higher ($p \leq 0.05$) than that induced by the corresponding cell-free filtrates. For instance, the MGI of *Fusarium* sp. AF68 significantly increased ($p \leq 0.05$) from 0 to 61.8% in the presence of *Pseudomonas* sp. BS95 cell cultures when compared to the respective cell-free filtrates (Figure 3). The results showed that *Bacillus* sp. BS36 was the PGPR strain with the best performance in suppressing the growth of the tested fungi. Therefore, experiments were carried out to induce and characterise the antifungal activity of the strain.

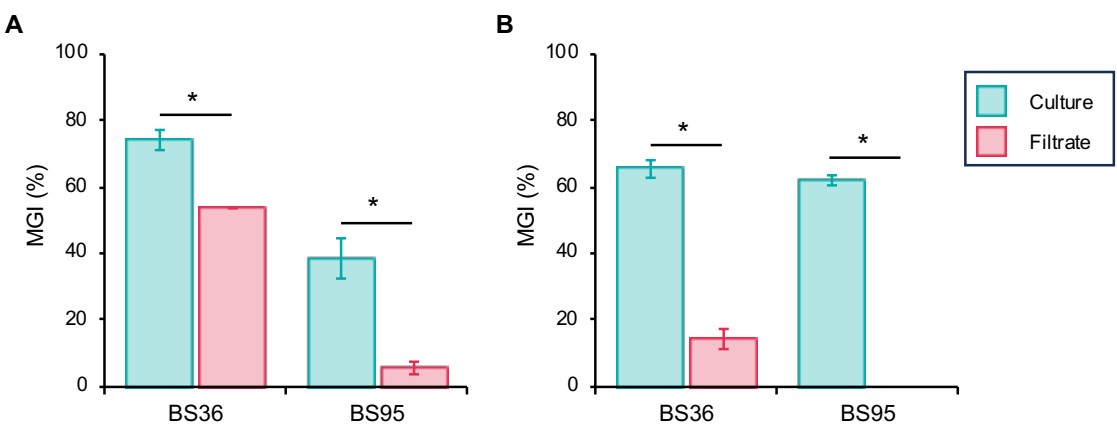

**Figure 3.** Antifungal activity of cultures and cell-free filtrates of *Bacillus* sp. BS36 and *Pseudomonas* sp. BS95 against *Alternaria* sp. AF12 (**A**) and *Fusarium* sp. AF68 (**B**), expressed by MGI. Values represent the mean $\pm$ SD (n = 4). Asterisks (*) indicate mean values significantly different ($p \leq 0.05$) according to independent samples *t*-test ((**A**): *t* (3) = 10.99 and $p < 0.001$ for *Bacillus* sp. BS36, and *t* (3.6) = 8.87 and $p < 0.001$ for *Pseudomonas* sp. BS95; (**B**): *t* (6) = 21.03 and $p < 0.001$ for *Bacillus* sp. BS36, and *t* (3) = 63.36 and $p < 0.001$ for *Pseudomonas* sp. BS95).

### 3.4. Bacterial Co-Culturing

The liquid co-cultivation of *Bacillus* sp. BS36, the strain with higher antifungal activity, with the remaining eight PGPR strains was evaluated through pairwise interactions. The cell-free filtrates of the co-cultures were tested for their antifungal activity against *Alternaria* sp. AF12 and *Fusarium* sp. AF68. The interspecific co-culture of *Bacillus* sp. BS36 and *Pseudomonas* sp. BS95 was the only one that exhibited improved antifungal activity. The MGI of *Fusarium* sp. AF68 induced by the cell-free filtrates of the co-cultures (20.5%) was significantly higher ($p \leq 0.05$) than those induced by the *Bacillus* sp. BS36 and *Pseudomonas* sp. BS95 monocultures' filtrates (14.4 and 0%, respectively). Yet, the high antifungal activity against *Alternaria* sp. AF12 detected for the *Bacillus* sp. BS36 monocultures' filtrates (53.9%) was not verified in the co-culture assay (30.7%) (Table 2).

**Table 2.** Antifungal activity of co-culture and monocultures filtrates of *Bacillus* sp. BS36 and *Pseudomonas* sp. BS95 against *Alternaria* sp. AF12 and *Fusarium* sp. AF68, expressed by MGI.

| Treatment [1] | Mycelial Growth Inhibition (MGI,%) [2] | |
|---|---|---|
| | *Alternaria* **sp. AF12** | *Fusarium* **sp. AF68** |
| BP | $30.8 \pm 0.0$ [b] | $20.2 \pm 1.7$ [a] |
| B | $53.8 \pm 0.0$ [a] | $14.4 \pm 3.2$ [b] |
| P | $5.8 \pm 1.9$ [c] | $0.0 \pm 0.0$ [c] |

[1] BP indicates co-culture of *Bacillus* sp. BS36 and *Pseudomonas* sp. BS95; B indicates monoculture of *Bacillus* sp. BS36; P indicates monoculture of *Pseudomonas* sp. BS95. [2] Values represent the mean $\pm$ SD (n = 4). Values represent the mean $\pm$ SD (n = 4). Different letters in the same column indicate mean values significantly different ($p \leq 0.05$) according to Tukey test (ANOVA, $F = 1366.98$ and $p < 0.001$ for *Alternaria* sp. AF12, and $F = 74.36$ and $p < 0.001$ for *Fusarium* sp. AF68).

### 3.5. Effect of Target Microorganism on Increasing Antifungal Activity

The MGI of *Alternaria* sp. AF12 and *Fusarium* sp. AF68 induced by *Bacillus* sp. BS36 changed with the addition of heat-inactivated cells of the target microorganisms (Table 3). The addition of inactivated cells of *Alternaria* sp. AF12 and *Fusarium* sp. AF68 significantly increased ($p \leq 0.05$) the growth inhibition of *Fusarium* sp. AF68. However, only the addition of inactivated cells of *Fusarium* sp. AF68 significantly increased ($p \leq 0.05$) the growth inhibition of *Alternaria* sp. AF12 (Table 3).

**Table 3.** Effects of the target microorganism on the antifungal activity of *Bacillus* sp. BS36 filtrates against *Alternaria* sp. AF12 and *Fusarium* sp. AF68, expressed by MGI.

| Treatment [1] | Mycelial Growth Inhibition (MGI,%) [2] | |
|---|---|---|
| | *Alternaria* **sp. AF12** | *Fusarium* **sp. AF68** |
| BA | $57.7 \pm 0.0$ [b] | $18.3 \pm 1.7$ [b] |
| BF | $63.5 \pm 1.9$ [a] | $36.5 \pm 5.8$ [a] |
| B | $59.6 \pm 1.9$ [b] | $7.7 \pm 3.8$ [c] |

[1] BA indicates co-culture of *Bacillus* sp. BS36 and inactivated *Alternaria* sp. AF12; BF indicates co-culture of *Bacillus* sp. BS36 and inactivated *Fusarium* sp. AF68; B indicates monoculture of *Bacillus* sp. BS36. [2] Values represent the mean $\pm$ SD (n = 4). Different letters in the same column indicate mean values significantly different ($p \leq 0.05$) according to Tukey test (ANOVA, $F = 10.42$ and $p = 0.005$ for *Alternaria* sp. AF12, and $F = 37.74$ and $p < 0.001$ for *Fusarium* sp. AF68).

### 3.6. Characterisation of Antifungal Metabolites Produced by Bacillus sp. BS36

In order to obtain an insight into the chemical nature of the antifungal metabolites, the *Bacillus* sp. BS36 cell-free filtrates were tested for their antifungal activity against *Alternaria* sp. AF12 and *Fusarium* sp. AF68 before and after physicochemical treatments. As shown in Figure 4, proteinase K digestion and heat treatment did not significantly ($p > 0.05$) affect the antifungal activity of the *Bacillus* sp. BS36 filtrates against *Fusarium* sp. AF68. However, heating the cell-free filtrates at 80 °C for 30 min significantly increased ($p \leq 0.05$) the growth inhibition of *Alternaria* sp. AF12 compared to the untreated filtrates (from 59.6 to 71.2%). The antifungal activity against *Alternaria* sp. AF12 produced by the *Bacillus* sp. BS36 filtrates was resistant to enzymatic digestion by proteinase K (Figure 4).

LPs were extracted from the cell-free filtrates of *Bacillus* sp. BS36 by acid precipitation and methanol extraction. The antifungal activity of the crude extracts of the LPs was tested against *Alternaria* sp. AF12 and *Fusarium* sp. AF68 and compared with that of the untreated filtrates. The extracts of the LPs inhibited the growth of both fungal strains (65.4% MGI for *Alternaria* sp. AF12, and 7.7% MGI for *Fusarium* sp. AF68), and there was no significant difference ($p > 0.05$) between the untreated and extracted filtrates (Figure 4).

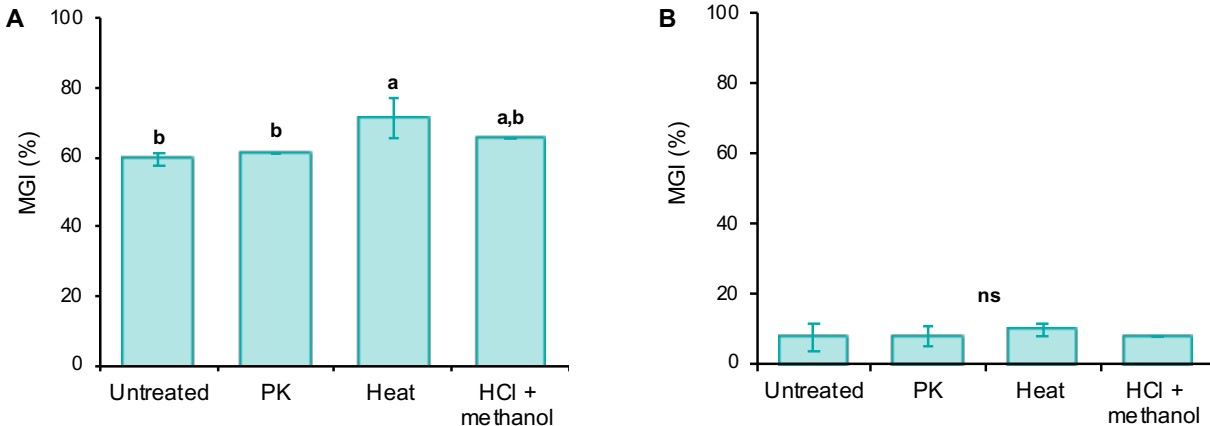

**Figure 4.** Effects of proteinase K, heat, and acidification (HCl) and methanol extraction on the antifungal activity of cell-free filtrates of *Bacillus* sp. BS36 against *Alternaria* sp. AF12 (**A**) and *Fusarium* sp. AF68 (**B**). Values represent the mean ± SD (n = 4). Bars with different letters indicate mean values significantly different ($p \leq 0.05$) according to Tukey test, and "ns" indicate not significant ($p > 0.05$) (ANOVA, (**A**): *F* = 8.42 and *p* = 0.003, (**B**): *F* = 0.43 and *p* = 0.735).

### 3.7. LPs Detection by FTICR-MS

The cell-free filtrates from *Bacillus* sp. BS36 were analysed with FTICR-MS. The FTICR-MS analysis revealed that fengycin- and surfactin-like LPs were produced by *Bacillus* sp. BS36. A peak with an *m/z* ratio of 1485.78515 corresponds to the mass of a [M + Na]$^+$ ion of fengycin-C16 ($C_{72}H_{110}N_{12}O_{20}$) with a molecular weight of 1462.79538 Da. A peak with an *m/z* ratio of 1074.64629 corresponds to the mass of a [M + K]$^+$ ion of surfactin-C15 ($C_{53}H_{93}N_7O_{13}$) with a molecular weight of 1035.68259 Da. No significant peaks for other LPs were detected in the FTICR-MS analysis.

### 4. Discussion

Our data suggest that PGPR, including some of selected strains of *Bacillus* and *Pseudomonas*, may also work as BCAs. Antagonism is a significant mechanism for suppressing pathogen growth through the production of antifungal compounds [46,47]. The use of antagonistic strains is one of the most important biological control technologies for disease management due to its environmental and biological safety.

A phylogenetic analysis of 16S rRNA showed the cluster of the PGPR strains under study with three different genera, namely *Bacillus*, *Priestia*, and *Pseudomonas*. The identification of the strains at the species level was not possible and would only be accomplished with additional analyses using other suitable DNA barcodes [48–50].

It was found that most *Bacillus* and *Pseudomonas* strains under study had direct antifungal activity against *Alternaria* sp. AF12 and/or *Fusarium* sp. AF68 using dual culture assays. These results are in line with previous studies reporting the high antifungal efficiency of several strains of *Bacillus* and *Pseudomonas* spp. against relevant phytopathogenic species of *Alternaria* and *Fusarium* [20,25,51–56]. In contrast, none of the *Priestia* strains showed antifungal activity against both fungi. *Bacillus* sp. BS36 and *Pseudomonas* sp. BS95 were the most efficient strains in inhibiting the growth of *Alternaria* sp. AF12 and *Fusarium* sp. AF68. Moreover, *Bacillus* sp. BS36 showed high antifungal activity against *Alternaria* sp. FP3, *Botrytis* sp. SM-D1, *Fusarium* sp. SM-D3, and *Stemphylium* sp. FP5, acting as a broad-spectrum antagonistic strain.

Cell-free filtrates of *Bacillus* sp. BS36 showed antifungal activity against both *Alternaria* sp. AF12 and *Fusarium* sp. AF68. These results suggest that this strain secreted antifungal metabolites that directly inhibited the phytopathogenic fungi. *Bacillus* spp. are known to produce a wide range of peptide and non-peptide antimicrobial compounds [26,57,58]. However, higher antifungal activity was recorded when cell cultures were used in dual culture assays, as also shown by Elshafie et al. [59]. As reviewed by Zhang et al. [60],

microorganisms can influence the environment and induce the production of specific metabolites by other microorganisms. Thus, the interactions between antagonistic bacteria and targeted fungi may stimulate the antagonistic bacteria to produce or over-express certain antifungal compounds.

Considering the importance of microbial interactions in the production of new antimicrobial compounds [61,62], *Bacillus* sp. BS36 was co-cultured with the other PGPR strains in an attempt to obtain cell-free filtrates enriched with new antifungal metabolites and, therefore, with increased antifungal activity. The co-culture of *Bacillus* sp. BS36 and *Pseudomonas* sp. BS95 showed improved antifungal activity against *Alternaria* sp. AF12. Wu et al. [63] showed that microbial interactions through the co-culturing of biocontrol microorganisms enhanced antifungal activity against *B. cinerea* due to the production of specific compounds. Li et al. [64] demonstrated similar effects against *F. graminearum.* Thus, the present results suggest that the interaction between *Bacillus* sp. BS36 and *Pseudomonas* sp. BS95 under co-culture conditions induces the production of specific antifungal compounds able to inhibit the growth of *Alternaria* sp. AF12. These cell-free filtrates obtained by bacterial co-culture would have an advantage in the production of BCAs.

Although there is evidence that co-culturing with heat-inactivated cells, cell-free filtrates, or extracts may not always be sufficient to induce the production of secondary metabolites [61], *Bacillus* sp. BS36 was also co-cultured with heat-inactivated cells of the target microorganisms, *Alternaria* sp. AF12 and *Fusarium* sp. AF68, in an attempt to obtain cell-free filtrates with increased antifungal activity. The co-culture of *Bacillus* sp. BS36 with heat-inactivated cells of *Fusarium* sp. AF68 showed improved antifungal activity against both fungi, while the co-culture of *Bacillus* sp. BS36 with heat-inactivated cells of *Alternaria* sp. AF12 showed improved antifungal activity only against *Fusarium* sp. AF68. It can be hypothesised that the secretion of antifungal metabolites by *Bacillus* sp. BS36 was induced by a stimulus from the target microorganism, which resulted in cell-free filtrates with increased antifungal activity.

Microbial interactions play a crucial role in shaping the dynamics of biological communities [65]. This study shows that these interactions can be exploited in vitro to enhance antagonism against phytopathogenic fungi, which is instrumental in the transition to more sustainable agriculture. Moreover, the study provides evidence that PGPR may also be efficient BCAs and that their biological control activity may be mediated through the production of antimicrobial metabolites. These metabolites' production can be enhanced through specific co-culturing conditions. Understanding the intricate microbial interactions and their potential to enhance antagonism against phytopathogenic fungi requires a holistic approach that considers the ecological and molecular aspects of these relationships.

The extracellular metabolites of *Bacillus* sp. BS36 had the same antifungal activity against *Alternaria* sp. AF12 and *Fusarium* sp. AF68, even after proteinase K treatment. However, antifungal activity against *Alternaria* sp. AF12 significantly increased after heating to 80 °C as compared to the untreated control filtrate. These findings suggest that the antifungal activity of cell-free filtrates against *Alternaria* sp. AF12 may be due, in part, to heat labile compounds. It can be hypothesised that the high temperature may have caused the degradation of certain compounds, resulting in the maturation of BCA or formation of new bioactive compounds with stronger antifungal properties. Alternatively, it may have affected metabolites that negatively interacted with the antimicrobial metabolites.

Moreover, the untreated cell-free filtrates had the same antifungal activity against *Alternaria* sp. AF12 and *Fusarium* sp. AF68 as the crude extracts of LPs. These results suggest that the LPs' secretion was involved in the in vitro inhibition of the growth of both fungi by *Bacillus* sp. BS36, as antifungal activity did not decrease in the presence of crude extracts of the LPs. LPs are secondary metabolites of particular interest due to their unique structure and bioactivity. LPs can be resistant to hydrolysis by peptidases and proteases and can withstand relatively high temperatures [66].

According to the FTICR-MS analysis, *Bacillus* sp. BS36 co-produced two types of LPs, fengycin and surfactin. The presence of these metabolites in the cell-free filtrates could

explain their high efficiency in suppressing the growth of the potentially phytopathogenic fungi. Fengycin and surfactin are produced by several *Bacillus* strains and have antagonistic activity against different microorganisms [22,67–70]. Fengycin is known for its high fungitoxicity, specifically against filamentous fungi [28,32,71]. Surfactin is not fungitoxic by itself, but it can contribute to indirect protection processes that involve the induced systemic resistance of the host plant [72–74]. Previous studies have described the co-production of fengycin and surfactin by *Bacillus* strains and their synergistic effects in the control of plant diseases [72]. The co-production of multiple families of LPs by *Bacillus* sp. BS36 is an interesting and potentially useful feature. Further characterisation of these molecules could be relevant in reducing the use of synthetic pesticides.

## 5. Conclusions

Our data show that the PGPR *Bacillus* sp. BS36 was the most effective strain in suppressing the growth of the two potentially phytopathogenic fungi, *Alternaria* sp. AF12 and *Fusarium* sp. AF68. Additionally, this strain exhibited strong antifungal activity against four other potentially phytopathogenic fungi, namely *Alternaria* sp. FP3, *Botrytis* sp. SM-D1, *Fusarium* sp. SM-D3, and *Stemphylium* sp. FP5. Although the cell cultures of *Bacillus* sp. BS36 inhibited fungal growth more than its cell-free filtrates, the latter also revealed high antifungal activity. It was shown that these cell-free filtrates contained fengycin- and surfactin-like lipopeptides, which may be responsible for its antifungal activity. This study also highlights the potential of microbial co-culturing to enhance the antifungal activity of *Bacillus* sp. BS36, and therefore supports that it should be further explored as a candidate to develop new BCA products.

To characterise *Bacillus* sp. BS36 as a potential BCA, it is essential to identify it at the species level. Therefore, additional molecular analyses using alternative primers/probes or long-read sequencing should be carried out. Evaluating the potential of PGPR to work as BCAs under field conditions is crucial to understanding how biotic and abiotic factors affect their activity and interactions. Moreover, the successful implementation of BCAs requires sustainable agricultural practices that promote the establishment and persistence of beneficial microorganisms in the plant ecosystem.

**Author Contributions:** Conceptualization, A.M.S., C.C. (Cristina Cruz) and L.C.; methodology, A.M.S., C.C. (Cristina Cruz) and L.C.; investigation, A.M.S., A.S. and J.M.; resources, C.C. (Cristina Cruz) and L.C.; data acquisition: A.M.S. and A.S.; data curation, A.M.S. and T.D.; statistical analysis, A.M.S. and L.C.; FTICR-MS analysis, C.C. (Carlos Cordeiro), M.S.S. and J.L.; writing—original draft preparation, A.M.S.; writing—review and editing, A.M.S., C.C. (Cristina Cruz), L.C. and T.D.; supervision, C.C. (Cristina Cruz) and L.C.; project administration, C.C. (Cristina Cruz) and L.C.; funding acquisition, C.C. (Cristina Cruz) and L.C. All authors have read and agreed to the published version of the manuscript.

**Funding:** This research was funded by (i) Portuguese funds from Fundação para a Ciência e a Tecnologia through the project UIDB/00329/2020 (DOI 10.54499/UIDB/00329/2020), the Researcher contract to T.D. (DOI 10.54499/DL57/2016/CP1479/CT0009), the PhD grant to J.L. (2023.05150.BDANA), the Portuguese Mass Spectrometry Network (LISBOA-01-0145-FEDER-022125) and the BioISI research centre (UIDB/04046/2020-DOI: 10.54499/UIDB/04046/2020 and UIDP/04046/2020-DOI: 10.54499/UIDP/04046/2020). We also acknowledge the support from the European project EU_FT-ICR_MS, funded by the European research and innovation programme Horizon 2020 (project no. 731077). The APC was funded by BioScale.

**Institutional Review Board Statement:** Not applicable.

**Informed Consent Statement:** Not applicable.

**Data Availability Statement:** The raw data supporting the conclusions of this article will be made available by the authors on request.

**Conflicts of Interest:** Authors Ana Soares and Luís Carvalho were employed by the company BioScale, Rua Nova da CEE. The remaining authors declare that the research was conducted in the absence of any commercial or financial relationships that could be construed as a potential conflict of interest.

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
