# Peer review of "Microbial Interactions as a Sustainable Tool for Enhancing PGPR Antagonism against Phytopathogenic Fungi"

_sustainability, doi:10.3390/su16052006_

Round 1
Reviewer 1 Report
Comments and Suggestions for Authors
- This paper correspond for scope of journal.
- The title corresponds to the content of the paper.
- This study represents a significant contribution to developing more safety method for protection plant against patthogens, i.e. developing biological control technologies for disease management based on antagonistic strain through production antifugal components (metabolites) due to its environmental and biological safety.
- The main question of manuscript is addressed to identify antifungal efficiency of three genera Bacillus, Priestia, and Pseudomonas to suppression Alternaria sp. and/ or Fusarium sp. using dual culture assays.
- The obtained result showed that Bacillus and Pseudomonas strains had direct antifungal activity against Alternaria sp. AF12 and/ or Fusarium sp. AF68 what is in line with previous studies reporting the high antifungal efficiency of several strains of Bacillus and Pseudomonas spp. against relevant phytopathogenic species of Alternaria and Fusarium.
- Biological control agents (BCAs). to provide solutions that can be more accessible and adoptable by the farmers and in the same time requires the development of sustainable agricultural practices that promote the establishment and persistence of beneficial microorganisms in the plant ecosystem.
- The aim of research is not clearly pointed out.
It is necessary on the end of chapter of Introduction write aim of investigation.
(It shoud be : The aim of investigation was establishing (i) microbial interactions through co-cultivation were explored to increase the production of existing and new metabolites to enhance the antifungal activity against phytopathogenic fungi, (ii) study of the nature and stability of extracellular metabolites with antimicrobial activity present in the obtained cell-free filtrates were determined and (iii). the presence of LPs in cell-free filtrates. )
- Key words are appropriate, but (Fourier-transform ion cyclotron resonance mass spectrometry should be use only spectrometry).
- For study used adequate methods.
- Results are clearly presented and discussed.
- Tables, figures, pictures are clear.
- The conclusions are not pointed out as a chapter of paper. Authors should be write particular chapter: Conclusion. The conclusions need derived on the base of obtained results!
- Manuscript is acceptable after minor corrections,!
Author Response
Response: Thank you for your review and your interesting comments and suggestions.
Regarding your first suggestion (i.e., the aim of research is not clearly pointed out), we agree with the Reviewer. Therefore, the aim of our study was clearly pointed out at the end of the introduction, as suggested.
Regarding the keywords, we thank you for your comment. Although the method we used (i.e., Fourier-transform ion cyclotron resonance mass spectrometry) is included in the broad term spectrometry, we consider that the technique we used is so specific and enables such a fine detection of compounds that it qualifies as a keyword. Therefore, we kept ‘Fourier-transform ion cyclotron resonance mass spectrometry’ as keyword.
Finally, we fully agree with the suggestion to include a conclusion section where we summarize and highlight the results obtained and point future perspectives.
Reviewer 2 Report
Comments and Suggestions for Authors
Dear authors!
Thank you for providing the article. You have done some interesting work using classical methods. Figure 2 is the highlight of the article. The article makes a positive impression.
main question addressed by the research: The use of rhizosphere bacteria, in particular Bacillus subtilis, to increase the resistance of cultivated plants to biotic stress caused by phytopathogenic fungi.
original or relevant for the field: The use of rhizosphere bacteria, namely the strain Bacilus sp. BS36 with another strain and PGPR to inhibit the growth of Alternaria sp. and Fusarium sp. Identification of surfactin and fengycin-like compounds among substances secreted by bacteria.
There is different information about the role of bacteria of the genus Bacillus in interaction with plants. Many researchers classify them as PGPR bacteria. The data available in this article support this opinion and explain the fungicidal effect by the synthesis of surfactin and fengycin-like compounds by bacteria.
There is no conclusion in the article. I recommend that authors add such a section to the article. Most of the references are relevant, but it is better to strengthen the reference list with references for the last 3 years. The drawings are made with high quality, there is statistical data processing.
Notes:
1) in the abstract you need to write more data obtained.
2) In the Introduction, indicate the purpose of the study.
3) It is necessary to add a Conclusion.
4) It is advisable to update the list of references by adding fresh references for the last 3 years.
Respectfully Yours, reviewer
February 01, 2024
Author Response
Response: Thank you for your review and your interesting comments and suggestions.
Regarding your first suggestion (changes in the Abstract), we fully agree and have added more data to make the Abstract more complete.
Regarding your opinion about the purpose of the study, we understand your comment based on the initial version of the manuscript. However, we consider that the full purpose and implications of our data were not clear. Therefore, the aim was clearly pointed out at the end of the Introduction and highlighted along the manuscript.
Further, we fully agree in including a Conclusion section. We conclude the study based on the obtained results and future prospects.
Finally, we agree in including more updated references and we included 5 new references ([14, 25, 50, 72, 73] for the last years (since 2020) in the list of references.
Reviewer 3 Report
Comments and Suggestions for Authors
Generally, the manuscript does not add anything to current knowledge and repeats previously reported findings.
The methods are appropriate, however, there are other primers/probes that could be used with PCR to narrow down the species of the Bacilli being used. This would help to make a slightly different contribution to literature. There is also the potential to use long read sequencing on a purified isolate to get bacterial identification to strain level. This is another way a contribution to knowledge could have been made. You need to create a unique focus to make your work stand out.
Some specific comments are below.
First line of the introduction says 'significant'. How significant? You should only be using this term if you've carried out statistic tests.
You need to give the correct company information for Eurofins.
Currently, the manuscript is too general and cannot add new knowledge to literature.
Author Response
Response: Thank you for your review and your interesting comments and suggestions.
Thank you for your warning. ‘Significant’ should not be used in that sentence which has no statistical reference, and it was deleted and replaced by ‘major’ (“Plant diseases cause major economic losses and …”). The correction requested for company information of Eurofins was made: “Eurofins Genomics (Germany)”.
Reviewer 4 Report
Comments and Suggestions for Authors
In general, I like the work that you did, and I appreciate your effort to understand and describe the bacterial interactions for enhancing antagonism against fungal strain tested. This finding is important and provides important results about the knowledge and futures perspectives to provide solutions in biocontrol of phytopathogenic fungi in agriculture. Almost all collected data analysis were performed correctly. Only one comment, in methodology 2.5 section describe the time of collect of cell-free filtrates during bacterial growth.
Author Response
Response: Thank you for your review and your interesting comments and suggestions.
Regarding your suggesting for including more details on section 2.5, we agree. Although all the information was already present in section 2.1, which explains the methodology used to culturing the bacterial strains for the antifungal assays, we clarified this point and added, in Section 2.5, the time of collect of cell-free filtrates as suggested (“after 24-48 h, as described in Section 2.1”).
Round 2
Reviewer 3 Report
Comments and Suggestions for Authors
Generally, this is an improvement. I have a question below that would help identify species names. You really need to describe your species as 'select species'.
Line 339 - Mentioning Pseudomonas and Bacillus in this way is too general. Make to specifically refer to the strains you are using.
Have you tried other primers to try and get closer to a species name? You could also do long read sequencing from a pure isolate.
Author Response
Thank you for your review and your interesting comments and suggestions. Thank you for your warning. We fully agree that, in line 339, mentioning Pseudomonas and Bacillus in that way is too general. Therefore, we have specified that we are referring to some of the strains selected in this study as suggested (“Our data suggest that PGPR, including some of selected strains of Bacillus and Pseudomonas, may also work as BCAs”). Regarding your question about using other primers, our future goal is to be able to identify the strains of interest at species level. To do this, we will use other primers or long-read sequencing in the future, as we mentioned in the Conclusion. However, at the moment we only have the phylogenetic analysis based on the 16S rRNA gene, which has allowed us to make a reliable identification at the genus level. Although we fully agree with the importance of identification at species level, as we mentioned in the Discussion, we believe that the phylogenetic reconstruction presented in this study is sufficient to complement the obtained results. After this initial screening and selection of the most promising strains, we will focus our research and consequently carry out a more in-depth molecular analysis of the strains of interest.